# Spatial Coupling of Carbon Sink Capacity with High-Quality Development Based on Exploitation and Protection Pattern

**Lin Wang** [1,2,3], **Junsan Zhao** [1,2,3,*], **Fengxia Li** [4] **and Guoping Chen** [1,2,3]

1 Faculty of Land Resources Engineering, Kunming University of Science and Technology, Kunming 650093, China
2 Key Laboratory of Geospatial, Information Integration Innovation for Smart Mines, Kunming 650093, China
3 Spatial Information Integration Technology of Natural Resources in Universities of Yunnan Province, Kunming 650211, China
4 College of Architecture, Xi'an University of Architecture and Technology, Xi'an 710055, China
* Correspondence: junsanzhao@netease.com

**Abstract:** The optimization of carbon sink capacity patterns is a scientific basis for promoting the strategic goal of "carbon neutrality". The study aims to reveal the spatial heterogeneity of carbon sink capacity and its relationship with high-quality economic development. A new spatial pattern and supporting system framework for carbon sink land-use (CSL) efficiency were developed in Yunnan Province, China. The coordination level and driving factors between carbon sink capacity and high-quality development were measured using a coupling coordination model and geographic detector model. The results show that the constructed index system can effectively identify the spatial distribution pattern of CSL efficiency at the county and district scale. The development stage of the coupling coordination degree shows a trend of "wide at both ends and narrow in the middle" and the characteristics of coupling and coordinated development type present an "inverted triangle" state. The coupling coordination relationship between carbon sink capacity and high-quality development depends mainly on the regional natural background conditions, economic development, and urbanization level. The findings provide a scientific basis for decision making in the development and protection of territorial space and offer a new perspective for government to facilitate ecological carbon sink capacity and promote high-quality development.

**Keywords:** ecological carbon sinks; sustainable economic growth; spatial structure; coupling coordination; factor detection

## 1. Introduction

Global terrestrial ecosystems play a crucial role in mitigating climate change by serving as carbon sinks [1]. Carbon sinks are natural systems that absorb atmospheric carbon dioxide through biological and physical processes, providing essential ecosystem services such as water purification, soil conservation, and biodiversity conservation [2,3]. In the last 60 years, natural terrestrial sinks have absorbed around a quarter of the emissions, offsetting carbon emissions from industrial activities and fossil fuel combustion [4]. China aims to achieve sustainable and environmentally friendly high-quality development in line with its "14th Five-Year Plan" [5]. Preserving and restoring natural carbon sinks can contribute to sustainable economic growth while mitigating climate change, thereby supporting this development [6]. However, it is essential to conduct economic activities in a way that minimizes environmental harm and promotes the efficient use of natural resources to achieve such development.

Currently, research on terrestrial carbon sinks is primarily focused on measuring carbon sinks using different methods [1,7,8], studying their sources, causes, and impacts [9–12], and simulating scenarios for their development and trends [13,14]. While there have been numerous studies on the land-use change in carbon sinks and their spatial distribution [15–17],

there is a lack of research on optimization space exploitation and conservation by using carbon sink resource efficiency as a logical starting point, especially when evaluating carbon sink capacity, efficiency, and potential. Some studies have focused on evaluating the carbon sequestration potential of different resource areas such as agroforestry [18], wetlands [19], grassland, forest, and cropland [20]. However, the land-use classification and efficiency for carbon sink as a whole have not been systematically evaluated. The evaluation model of carbon sink capacity is insufficient, and the classification of CSL and their functional weight is ignored. In terms of high-quality economic development, scholars have mainly focused on researching the evaluation system and influencing factors [21,22]. These studies have enriched the research content of the evaluation index system. Some recent studies [23–25] have explored the relationship between carbon sequestration and economic growth, while others have constructed an indicator system based on the ecological niche theory to measure and analyze the ecological niche for high-quality development in resource-based cities. However, the coupling relationship between high-quality economic development and ecological carbon sinks has not been extensively explored.

Therefore, this paper aims to construct a framework for the space system of efficiency in land-use carbon sinks based on the identification of CSL. This framework will help reveal the spatial heterogeneity of carbon sink capacity and high-quality economic development using a coupling coordination model and geographic detector model. The study will also combine grid and county scales to provide a research foundation for development strategy, planning, and application.

## 2. Theoretical Framework

### 2.1. The Logical Relationship between Carbon Sink Capacity and High-Quality Economic Development

China's decision to achieve peak carbon neutrality is a crucial strategic move, taking into account domestic and international situations. Terrestrial ecosystems are recognized worldwide as a cost-effective means of reducing emissions, and "carbon sinks" play a critical role in this regard. By enhancing carbon sequestration capacity, we can maximize carbon fixation efficiency and derive additional ecological and economic benefits from these sinks. Moreover, achieving high-quality development requires reducing resource and energy consumption, environmental costs, and boosting productivity. A healthy ecological environment is also crucial to achieving this goal. Hence, both carbon sink capacity and high-quality development aim to address the pressing issue of resource and environmental constraints.

### 2.2. System Construction of Carbon Sink Capacity and High-Quality Economic Development

Carbon sink capacity is a significant endogenous driving force of carbon sequestration. The distribution pattern of carbon is influenced by land-use and land-cover change, and structural and environmental factors [23,24]. Improving carbon sink capacity requires a holistic approach that considers multiple land types. The carbon sink system is based on the eco-space as the primary source. The eco-space includes a continuous and complete landscape pattern, wetland and river systems, as well as living and productive spaces that function as carbon sinks [26,27]. Therefore, the carbon sink capacity system is constructed from three dimensions: CSL efficiency, carbon storage potential, and the ecological advantage of carbon sinks.

Furthermore, high-quality development has profound economic, social, and ecological implications. The evaluation system can be constructed from three aspects: economic growth closely linked to land use and exploitation efficiency, environmental sustainability, and social equity.

The methodology flow chart is presented in Figure 1. This study aims to quantitatively identify carbon stocks and sequestration, analyze evolutionary characteristics from 1980 to 2020, and comprehensively evaluate CSL effectiveness. Next, we will carry out a coupling coordination degree evaluation and divide it into different stages and types of development.

Finally, we will combine single-factor and two-factor analyses to understand the influence mechanisms of the carbon sink capacity system, high-quality development, and coupling degree of coordination.

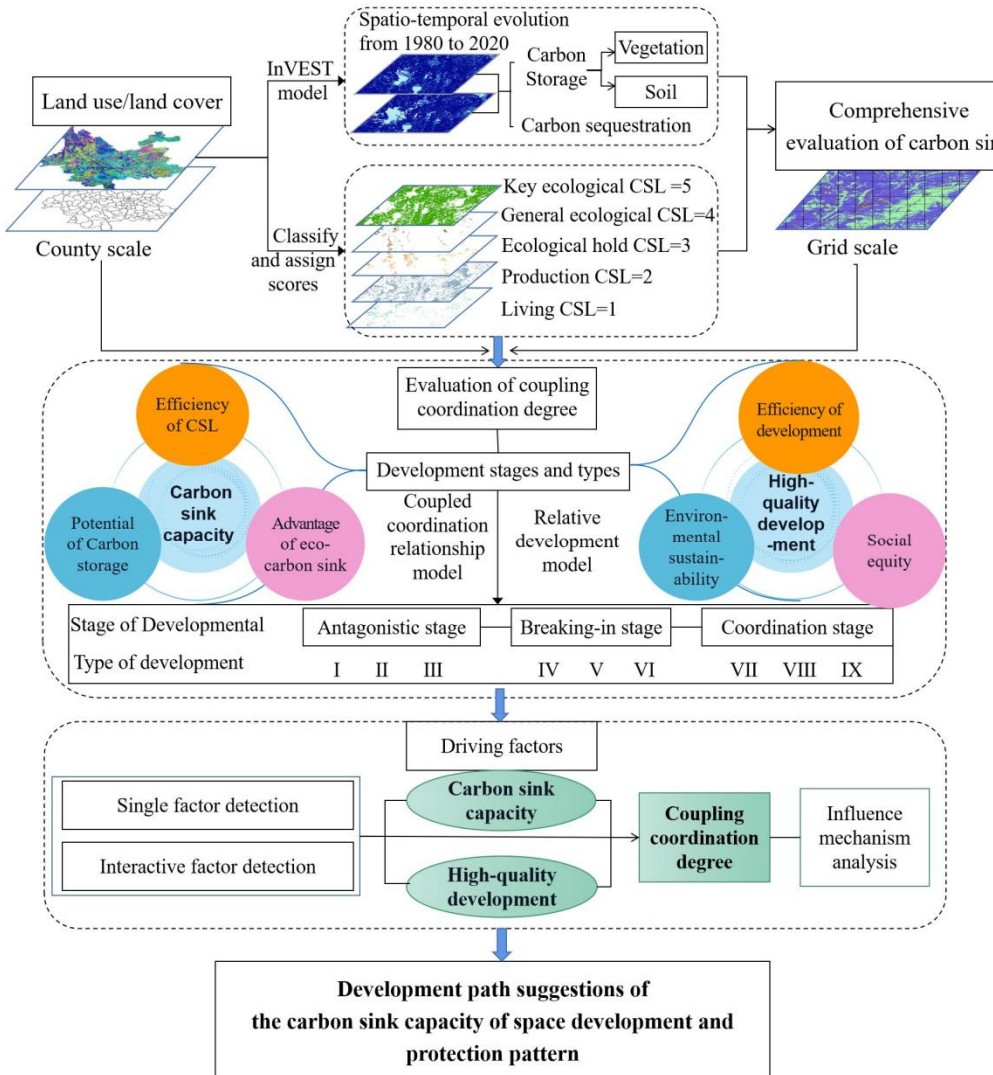

**Figure 1.** Theoretical framework of carbon sink capacity with high-quality development.

## 3. General Situation and Data Sources of the Study Area

### 3.1. Study Area

Yunnan Province (97°31′~106°11′ E, 21°8′~29°15′ N) is located in southwest China, in the upper reaches of the Yangtze River (Figure 2), with proximity to Myanmar, Laos, and Vietnam. The province spans an area of 3,834,000 km$^2$ and comprises 129 counties (cities and districts). Yunnan serves as China's window and gateway to Southeast Asia and South Asia and is a significant component of major national development strategies, such as the "Belt and Road Initiative" and the development of the Yangtze River Economic Belt. The landform in Yunnan Province is mainly mountainous and constitutes the primary part of the Yunnan–Guizhou Plateau in southwest China. The altitude range of 1000–3500 m accounts for 87.21% of the province's total land area, with the mountain region accounting for 88.64% of the total land area, and the area with a slope of more than 25° accounting for 43.54%. The total forest coverage rate in the province was 65.04% by 2021.

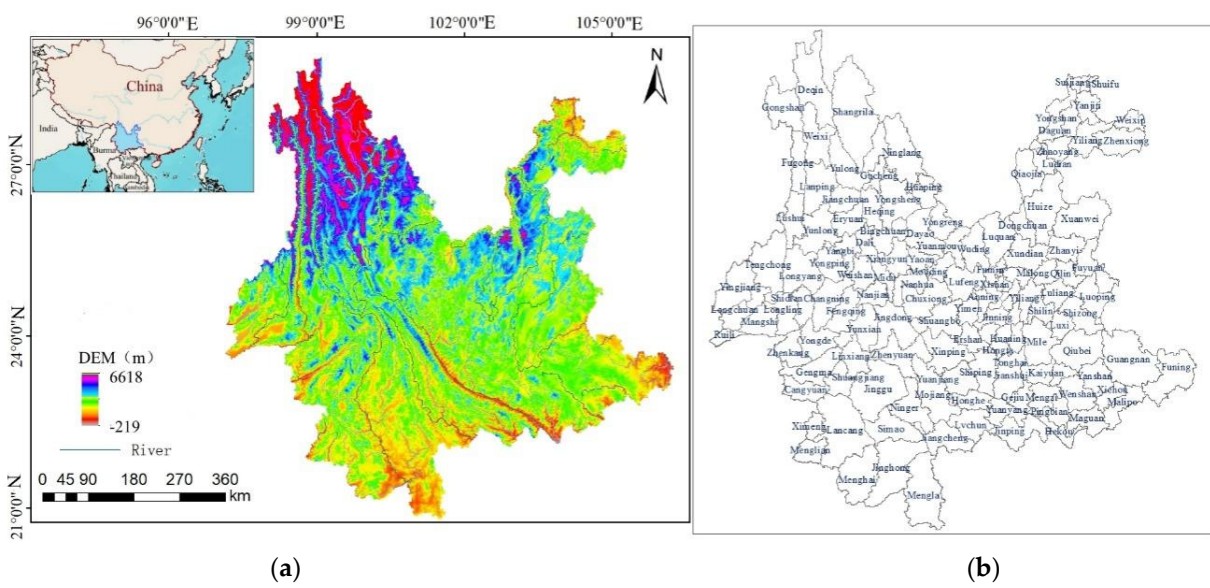

**Figure 2.** Study area: (**a**) location and topography of Yunnan province; (**b**) 128 countries and districts in Yunnan province.

## 3.2. Data Sources and Processing

The research data sets mainly include a satellite image data set, land-use/-cover data set, meteorological data set, and other related auxiliary data sets. Yunnan-wide land use data for 1980, 1990, 2000, 2010, and 2020 used in this study were obtained from the Chinese Academy of Sciences' Data Center for Resources and Environmental Sciences (http://www.resdc.cn, accessed on 18 May 2022), the land-use types are divided into 6 primary categories and 66 secondary categories. The elevation data come from the geospatial data cloud (http://www.gscloud.cn, accessed on 5 July 2022). The vector distribution data of rivers, roads, and towns come from the China National Surveying and Mapping Center (NGCC) (http://ngcc.sbsm.gov.cn, accessed on 5 July 2022). The soil data come from the World Soil Database (HWSD) (http://westdc.westgis.ac.cn/, accessed on 23 August 2022). Population, GDP, night light, and climate data all come from CAS (www.resdc.cn/data.aspx, accessed on 5 July 2022). Distance variables and road network density were extracted from OpenStreetMap (https://www.openstreetmap.org/, accessed on 11 September 2021). Other socio-economic data are from the Yunnan Statistical Yearbook. In this study, the spatial data were uniformly transformed into the WGS_1984_UTM_48N coordinate system. The data were vectorized and the basic evaluation database of "CSL efficiency" with a 5 km × 5 km grid was established.

## 4. Research Methods

### 4.1. Carbon Storage Identification Based on InVEST Model

The InVEST model is used to simulate changes in ecosystem services and includes a series of sub-models [28]. It can well evaluate the influence of LUCC on carbon storage changes, which has high practical significance in domestic and foreign research [29,30]. The InVEST carbon storage and sequestration model was used to estimate the total carbon storage in the study area and quantify sequestration over time [31], and it is calculated as follows:

$$C_i = C_{above} + C_{below} + C_{soil} + C_{dead}, \tag{1}$$

$$C_{total} = \sum_{i=1}^{n} A_i \times C_i, (i = 1, 2, \cdots, n), \tag{2}$$

$$C_{sequestration} = C_{total(t+1)} - C_{total(t)} \tag{3}$$

where $C_i$ is the total carbon density of the *i*-th land-use type; $C_{above}$ is the carbon density of aboveground biological soil organic carbon; $C_{below}$ is the carbon density of underground biomass; $C_{soil}$ is the carbon density of soil organic matter. $C_{dead}$ is the carbon density of dead organic matter. $C_{total}$ is the total carbon storage of the ecosystem. $C_{sequestration}$ is carbon sequestration [32]; $A_i$ is the area of Class i land-cover type; *n* is the number of land-use types, which was set as 6 in this study. The InVEST model was used to estimate carbon stocks in 1980, 1990, 2000, 2010, and 2020, and carbon sequestered from 1980 to 2020. We collected the carbon density of each land-use and land-cover type from the relevant literature (Table 1).

**Table 1.** Carbon density of each land cover type in the InVEST model.

| Land-Use/Land-Cover Type | $C_{above}$ | $C_{below}$ | $C_{soil}$ | $C_{dead}$ |
|---|---|---|---|---|
| Cultivated land | 7.99 | 3.4 | 64.18 | 0.19 |
| Woodland | 35.2 | 12.39 | 81.85 | 2.56 |
| Grassland | 7 | 12.76 | 74.56 | 3.73 |
| Water | 5.11 | 5.81 | 11.26 | 0.05 |
| Urban and rural built-land | 4.25 | 2.98 | 13.68 | 0.1 |
| Unused land | 2.51 | 0.62 | 12.94 | 0.2 |

*4.2. Evaluation System and Calculation of Comprehensive Degree Index of Carbon Sink Land*

4.2.1. Land-Use Classification of Carbon Sinks

Land-cover type determines the spatial variability of carbon sink [13]. First, the implementation plan to coordinate with the existing spatial pattern and functional zoning must be ensured, so as to identify the carbon sink space of the naturally ecological space, production space, and living space. By referring to the existing research on the spatial classification of ecological space, production space, and living space, the paper classifies the space of CSL, and then establishes the function index system of CSL (Table 2). Among them, ecological land was divided into key carbon sink ecological land, general carbon sink ecological land, and ecological carbon sink hold land, followed by production CSL and living CSL.

**Table 2.** Comprehensive degree index evaluation system of CSL.

| Main Categories | Secondary Categories | Function Scores of Carbon Sink | Land Use/land Cover Type (Primary) | Land Use/Land Cover Type (Secondary) | Value Orientation |
|---|---|---|---|---|---|
| Ecological land | Key carbon sink ecological land | 5 | Woodland | Shrubbery land Open woodland Forest land | Ecological service value + carbon sink trading value |
| | | | Grassland | High-coverage grassland | |
| | | | Unused land | Marsh land | |
| | General carbon sink ecological land | 4 | Woodland | Other Woodland | Ecological service value |
| | | | Grassland | Moderate coverage grassland Low-coverage grassland | |
| | | | Water | River | |
| | Ecological carbon sink hold land | 3 | Water | Canals, reservoirs, ponds, tidal flats, permanent glaciers and snowfields | Ecological hold value |
| | | | Unused land | Sand, saline-alkali land, bare land, Gobi, Alpine desert, Tundra paddy fields | |
| production space | Production carbon sink land | 2 | Cultivated land | Dry land | Carbon neutrality value in agriculture |
| living space | Living carbon sink land. | 1 | Urban and rural, industrial and mining, residential land | Country residential Other construction land | Urban and rural ecological value |

Scores must be assigned to carbon sinks according to the value of CSL [28]. Ecological land for key carbon sinks has the highest carbon storage, and has both ecological service value and carbon sink trading value, therefore, it has the highest score of 5 points [33]. With the decrease in carbon storage and value, the carbon sink function was successively decreased.

4.2.2. Comprehensive Efficiency Index of Carbon Sequestration of Land Cover

In order to reflect the extent and measure the capacity of regional CSL, the comprehensive index of CSL efficiency was calculated which combined with the comprehensive index method. Referring to the previous methods of the land-use comprehensive degree index [34], for this study, the carbon fixation degree of CSL was divided into five capacity indices, with higher indices indicating greater ecological carbon fixation. It is calculated as follows:

$$L = 100 \times \sum_{i=0}^{n} (A_i \times C_i) \tag{4}$$

where $A_i$ is the classification index of the carbon fixation degree of CSL. According to the carbon fixation function, the key ecological CSL = 5, general ecological CSL = 4, ecological hold CSL = 3, production CSL = 2, and living CSL = 1. $C_i$ is the area proportion of the corresponding land-use classification. The area of Yunnan was divided into a 5 km $\times$ 5 km grid for calculation.

*4.3. Construction of Coupling Model between Carbon Sink Capacity and High-Quality Development*

4.3.1. Evaluation Index Construction

The carbon sink capacity system is constructed from three dimensions: efficiency of CSL, potential of carbon storage, and advantage of ecological carbon sink. High-quality development is an important driving force for the evolution of spatial carbon sinks, in which the system is constructed from land use and exploitation efficiency, environmental sustainability, and social equity (Table 3). The entropy method (EWM) was used to determine the weight of each index according to the information amount of each index, and then the comprehensive evaluation was realized [35].

**Table 3.** Comprehensive index system and weight of spatial carbon sink capacity and high-quality development.

| Primary System | Subsystem (Index Weight) | Indicator Layer | Unit | Mean | Nature | Index Weight |
|---|---|---|---|---|---|---|
| Carbon sink capacity | Efficiency of CSL 0.374 | Carbon fixation degree of CSL | \ | Formula (4) | + | 0.374 |
| | Potential of Carbon storage 0.291 | Total carbon storage | % | Total area carbon storage | + | 0.002 |
| | | Proportion of added value of carbon storage | % | Increase in carbon storage /total carbon storage in the region | + | 0.037 |
| | | Proportion of carbon storage reduction value | % | Reduction in carbon storage/total carbon storage in the region | - | 0.103 |
| | | Average land carbon storage density | % | Carbon storage/total area | + | 0.108 |
| | Advantage of ecological carbon sink 0.335 | Ecological dominance of carbon storage | % | High-carbon storage land area/total area | + | 0.142 |
| | | Per capita ecological CSL area | km$^2$ | Ecological CSL area/total population | - | 0.218 |

**Table 3.** *Cont.*

| Primary System | Subsystem (Index Weight) | Indicator Layer | Unit | Mean | Nature | Index Weight |
|---|---|---|---|---|---|---|
| High-quality development | Efficiency of land use and exploitation 0.562 | Spatial exploitation intensity | km$^2$ | Construction land area/total area | - | 0.131 |
| | | Rate of increase in fixed capital investment | % | [(regional fixed assets investment in the current year/fixed assets investment in the previous year) − 1] × 100 | + | 0.153 |
| | | Output value of construction land per unit | Yuan | Total GDP/construction land area | + | 0.085 |
| | | Carbon intensity | t | Carbon emission quantity (carbon stock reduction)/gross regional product | - | 0.096 |
| | | Proportion of total output value of agriculture, forestry, animal husbandry and fishery | million yuan | Gross output value of agriculture, forestry, animal husbandry and fishery/gross regional product | + | 0.021 |
| | | Proportion of value added of primary industry | % | Value added of primary industry/gross regional product | + | 0.022 |
| | | Proportion of added value of secondary and tertiary industries | % | Added value of secondary and tertiary industries/gross regional product | + | 0.054 |
| | Environmental sustainability 0.272 | Forest coverage rate | % | Forest area/total area | + | 0.169 |
| | | Water area coverage | % | Water area/total area | + | 0.061 |
| | | Grassland coverage | % | Grassland area/total area | + | 0.042 |
| | Social equity 0.166 | Urban–rural income ratio | % | Per capita disposable income of rural residents/per capita disposable income of urban residents | + | 0.021 |
| | | Per capita revenue and expenditure ratio of local general public budget | Yuan | Per capita local general public budget revenue/per capita local general public budget expenditure | + | 0.063 |
| | | Per capita savings deposit level | Yuan per capita | savings surplus/Regional total population | + | 0.082 |

### 4.3.2. Construction of Coupling Coordination Relationship Model

The coupled coordination model describes the relationship between different systems at high and low levels of mutual promotion or restriction through the degree of coordinated development among different systems [36]. The degree of coupling coordination not only determines the structure and sequence of key subsystems, but also determines the development trend of the system from disorder to order. After standardized processing by

the step difference method, the coupling degree model was calculated, and the formula is as follows:

$$C_{ab} = \left\{ \frac{Z_a Z_b}{((Z_a + Z_b)/2)^2} \right\}^k \tag{5}$$

where $C_{ab}$ represents the coupling degree value of carbon sink capacity and high-quality development system, and the value interval is [0, 1]. The larger $C$ is, the stronger the system interaction is; $Z_a$ and $Z_b$ are comprehensive indexes of carbon sink capacity and high-quality development, respectively. $k$ is the adjustment coefficient ($k \geq 2$), and $k = 2$ was adopted in this paper.

In order to avoid obtaining pseudo evaluation results when different systems use small values at the same time, a coupling coordination degree model was introduced, as to accurately reflect whether the systems are mutually reinforcing at a higher level or closely linked at a lower level. The formula for calculating coupling coordination degree is as follows:

$$D_{ab} = (C_{ab} \times T_{ab})^{1/2} \tag{6}$$

$$T_{ab} = \alpha Z_a + \beta Z_b \tag{7}$$

where $D_{ab}$ is the degree of coordination, and the value interval is [0, 1]. The larger the value is, the better the consistency is. $\alpha$, $\beta$ are undetermined coefficients. In this paper, $\alpha$, $\beta$ are 0.5.

### 4.3.3. Relative Development Model

The coupling coordination model can accurately evaluate the development level of coupling coordination, but it is difficult to evaluate their relative development. To further measure the relative development degree between carbon sink capacity and high-quality development system, the relative development degree coefficient was calculated by the formula as follows:

$$P = Z_a / Z_b \tag{8}$$

where $P$ is the relative development degree. Referring to the existing research results [37], when $P > 1.2$, the carbon sink function is advanced. When $0.8 < P \leq 1.2$, the carbon sink function and high-quality economic development develop synchronously. When $P \leq 0.8$, the carbon sink function lags behind.

### 4.4. Geographic Detector Model

The geographic detector model quantitatively expresses the spatial stratified heterogeneity (spatial heterogeneity) of the research object by analyzing the differences and similarities between the intra-layer variance and the inter-layer variance. Its calculation formula is as follows:

$$q = 1 - \frac{1}{N\sigma^2} \sum_{h=1}^{L} N_n \sigma_h^2 \tag{9}$$

where $q$ is the spatial heterogeneity of carbon sink capacity system, high-quality development system, and the coupling coordination relationship between them; $N$ is the total number of samples in the group city region; $\sigma^2$ is the variance of the index; $h = 1, 2 \ldots, L$, $h$ indicates the partition and $L$ indicates the number of partitions. $q \in [0, 1]$. The size of $q$ reflects the degree of spatial differentiation. The larger the value of $q$, the stronger the heterogeneity of spatial stratification; otherwise, the stronger the randomness of spatial distribution [38].

This paper explored the influencing factors of the coupling and coordination degree between carbon sink capacity and high-quality development in Yunnan Province from 12 aspects: the altitude (X1), slope (X2), air temperature (X3), soil type (X4), regional population (X5), afforestation area (X6), green space coverage (X7), water conservancy

facilities land (X8), river network density (X9), road network density (X10), transportation land (X11), and night light index (X12), among which, Nocturnal light data were analyzed as a factor of urban expansion as it could explain human activities.

## 5. Result and Analysis

### 5.1. Temporal and Spatial Evolution Characteristics of Carbon Storage and Carbon Sequestration

5.1.1. Temporal and Spatial Evolution Characteristics of Carbon Storage and Carbon Sequestration

Between 1980 and 2020, the overall trend of carbon storage in Yunnan Province was decreasing, as shown in Figure 3a. Specifically, carbon storage declined from $4306.453 \times 10^6$ t in 1980 to $4274.301 \times 10^6$ t in 2020. The rate of reduction was slower between 1980 and 2010, carbon storage showed a slight increase, but from 2010 to 2020, the decline rate accelerated, and the rate of change reached 0.61%.

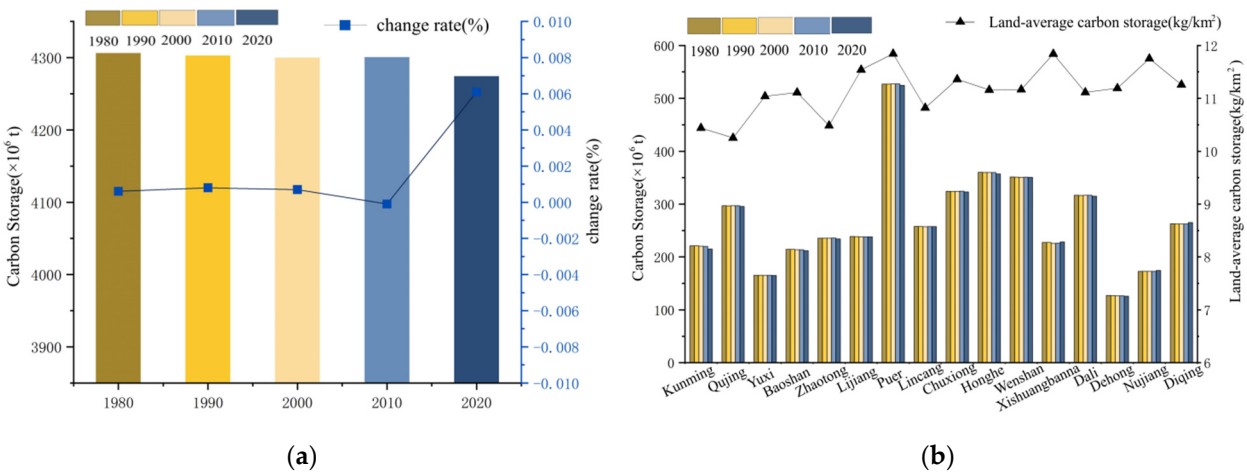

(**a**)  (**b**)

**Figure 3.** Changes in carbon storage in Yunnan from 1980 to 2020: (**a**) the trend of carbon storage in Yunnan Province; (**b**) the distribution of carbon storage among different regions.

The distribution of carbon storage varied significantly across different regions, as shown in Figure 3b. In proportion to regional area, Pu'er had the most prominent carbon reserve advantage. The regions with higher carbon reserves were Honghe, Wenshan, Chuxiong, and Dali, while Dehong, Nujiang, and Yuxi City had lower carbon reserves. However, it is worth noting that Pu'er City, Xishuangbanna, and Nujiang had the highest land-average carbon storage, whereas Qujing, Kunming, and Zhaotong had the lowest land-average carbon storage.

Combining vegetation carbon storage and soil carbon storage (Figure 4a,b), Yunnan Province's overall carbon storage exhibits spatial distribution characteristics of being "higher in the northwest, central and eastern regions, and lower in central and eastern regions". The areas with high-carbon storage values were mainly concentrated in mountainous forest areas, with a strip-like distribution, while areas with low-carbon storage values were widely distributed in urban and agricultural planting areas, with point-like strip distribution. To further characterize the local spatial characteristics of carbon storage evolution in Yunnan Province, carbon storage was divided into four levels by natural breakpoint method, with the proportion of high-carbon storage values being 57%. The proportion of higher carbon stocks increased from 23% in 1980 to 40% in 2020. The highest level of carbon storage, i.e., high-carbon storage area, was extracted to quantify the local spatial characteristics of land-use carbon emissions using the standard deviation ellipse method, as shown in Figure 4c. The long semi-axis of the ellipse represents the distribution direction of the carbon emission data, while the short semi-axis represents the distribution range of the data. The maximum area of the standard deviation ellipse in 1980 indicates the widest range of carbon storage, and the area of the standard deviation ellipse will decrease

in 2020. By comparing the standard deviation ellipse between 1980 and 2020, the center rotates about 3.19° to the northwest in 2020, and the long axis increases while the short axis decreases, indicating a decrease in carbon storage during this period.

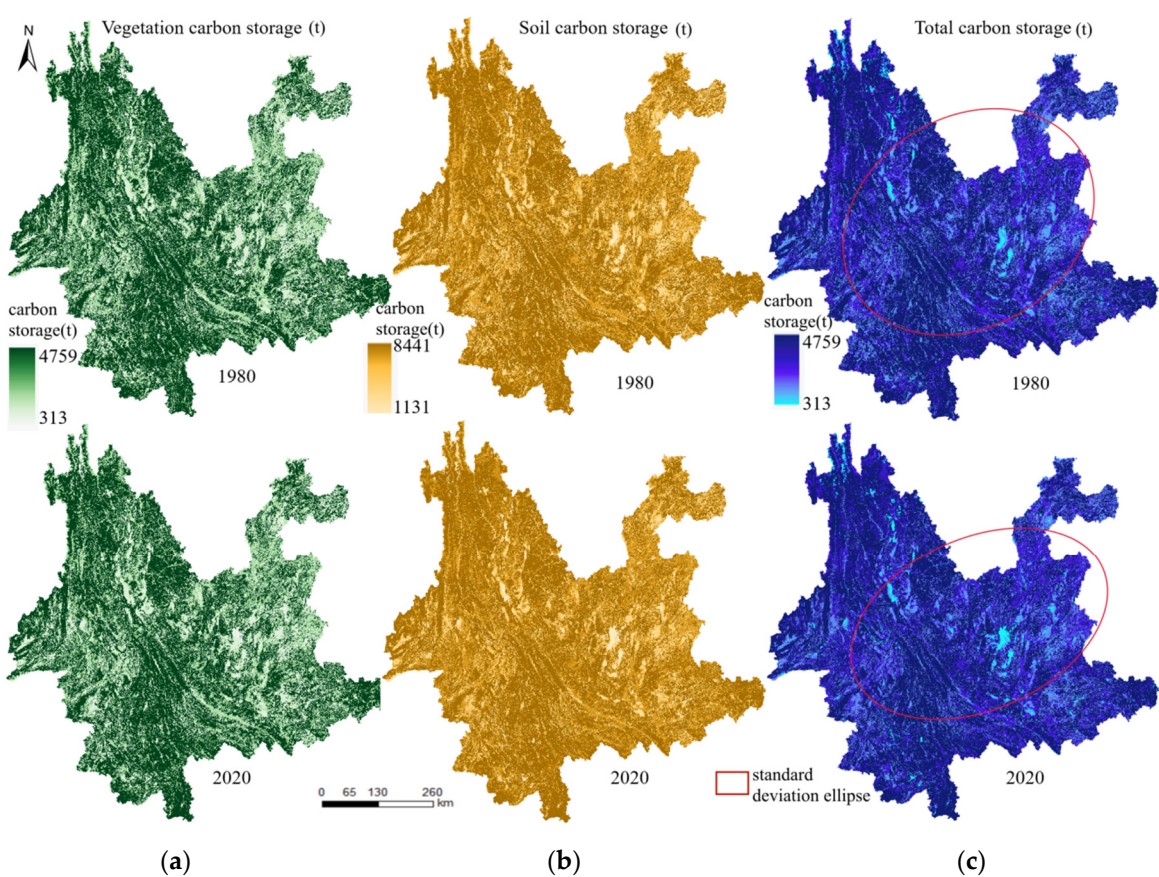

**Figure 4.** Spatial distribution and change in carbon storage in Yunnan from 1980 to 2020: (**a**) vegetation carbon storage; (**b**) soil carbon storage; (**c**) total carbon storage.

5.1.2. Temporal and Spatial Characteristics of Ecological Carbon Sinks

The analysis of the temporal and spatial characteristics of carbon sequestration rate caused by changes in land structure, and the change in carbon storage in Yunnan Province over the 40 year period, from 1980 to 2020, is shown in Figure 5a. A positive value represents an increase in carbon storage, which can be regarded as sequestered carbon, while a negative value represents a decrease in carbon storage, which can be regarded as emitted carbon. The analysis shows that (1) The carbon storage increased by $65,639 \times 10^6$ t over the past 40 years, and the carbon sequestration accounted for 16.35% of the regional area. The higher carbon sequestration value was mainly distributed in the northeast of Deqing, the southeast of Lincang, the west of Xishuangbanna and Pu'er, and the middle of Wenshan. (2) The reduced carbon storage was $318,017 \times 10^6$ t, accounting for 83.02% of the regional area. The reduced carbon storage area was mainly spread gradually around prefectural-level cities.

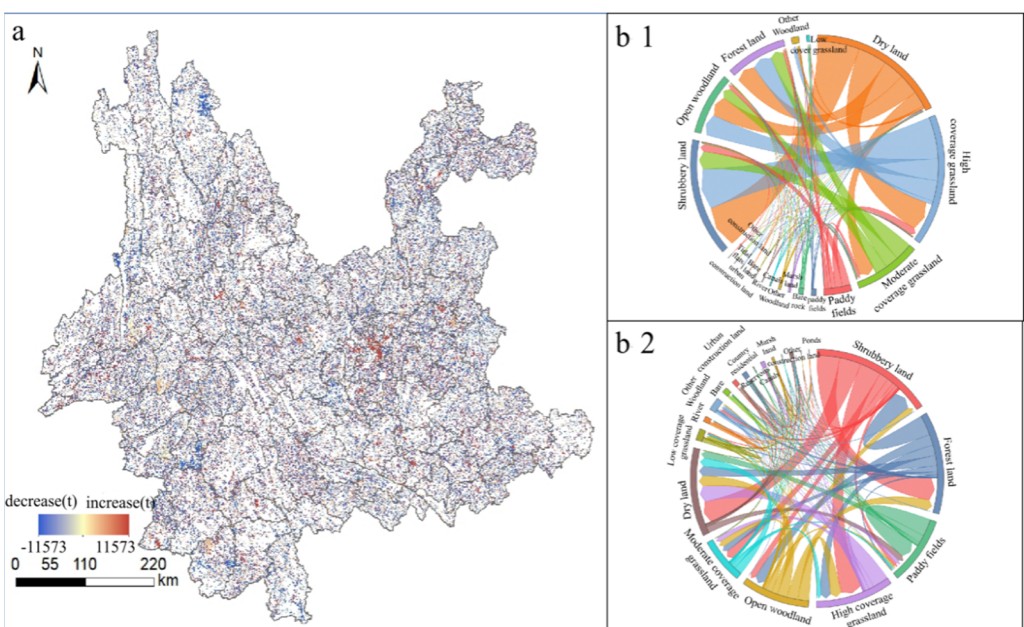

**Figure 5.** Schematic diagram of carbon storage changes during 1980–2020: (**a**) distribution of carbon storage increase and decrease: the left side of the legend is a decrease, and the right side is an increase; (**b1**) Land use transfer with increased carbon sinks in Yunnan Province; (**b2**) Land use transfer with reduced carbon sinks.

To further analyze land-use changes that form carbon sequestration, and the quantitative conversion relationship of land-use types for carbon sequestration, a chord diagram visualization was used, as shown in Figure 5(b1). The increase in carbon storage from 1980 to 2020 is mainly based on the transformation of the following land classes: the outflow of dry land, paddy land, and high-cover grassland shows considerable dominance, which mainly changed to shrub, open woodland, and forested land. The changes from village settlements and bare rock to medium cover grassland and medium cover grassland also had obvious effects on the increase in carbon storage. Secondly, some rural residential areas, other construction land, lakes and beaches have been transformed into dry land and paddy field. Figure 5(b2) shows that in the land-use change with carbon storage reduction, the transformation from woodland to grassland, grassland to dryland, and farmland to construction land are all common forms for land-use change with carbon storage reduction.

Combined with the evolution of the comprehensive index of CSL efficiency, Figure 6 shows regional differences in different dimensions from 1980 to 2020.

(1) In general, the comprehensive index of CSL efficiency is relatively high in the areas above middle and high altitudes, with the average altitude of most areas concentrated in the range of 1250–2250 m. Additionally, the comprehensive index of CSL efficiency is relatively higher in the southwest, south, southeast, and northeast, indicating that the carbon sequestration efficiency of the land-use ecosystem in these regions is higher. The areas with a low value for the comprehensive index of CSL efficiency are mainly located in the provincial capital Kunming, followed by Malone and Qilin of Qujing, Luxi and Hekou of Honghe, Xichou of Wenshan, Muding and Midu of Dali, Lianghe and Ruili of Baoshan, and Ludian, Suijiang, Shuifu and Weixin of Zhaotong. This illustrates that it is difficult to guarantee the quantity and quality of the "ecological space" in these areas.

(2) Regarding the trend of change, the CSL efficiency index in the counties with higher average altitude showed an increasing trend from 1980 to 2020, such as in Wenshan, Shuangbai, Panlong District, and Mengshi, while it decreased in Xichou, Lianghe, Gejiu, Pingbian, Hekou, and Ruili. The most significant change is that the CSL efficiency index in the central Yunnan urban agglomeration shows a trend of concentrated shrinkage.

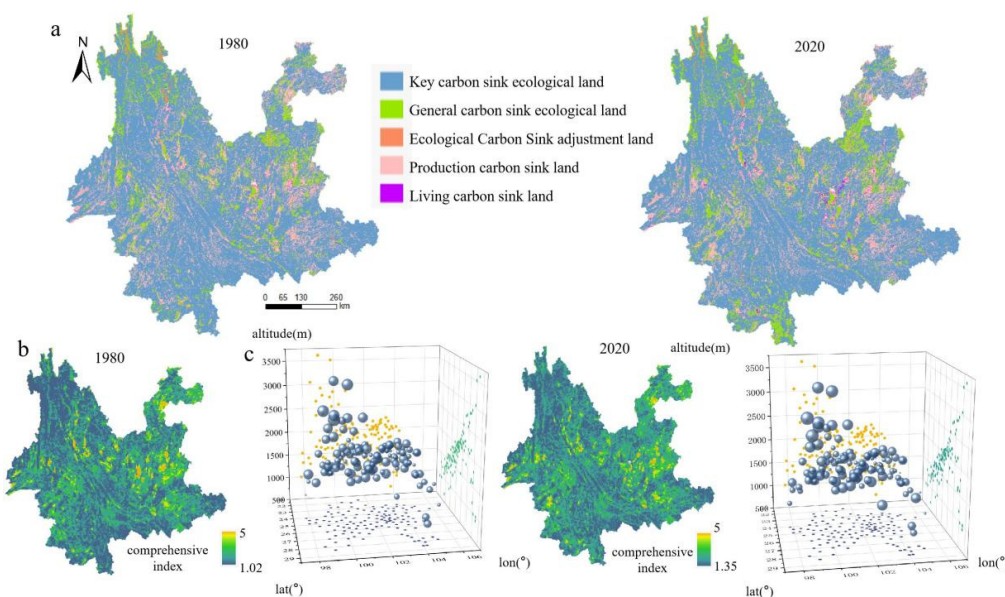

**Figure 6.** Classification of CSL, distribution, and change in comprehensive index of CSL efficiency: (**a**) land-use classification of carbon sinks; (**b**) distribution of comprehensive index of CSL efficiency in 1980 and 2020; (**c**) change in comprehensive index of CSL efficiency between 1980 and 2020.

*5.2. Comprehensive Function Evaluation and Coupling Coordination Analysis*

5.2.1. Spatial Pattern of Carbon Sink Capacity and Comprehensive Index of High-Quality Development Level

The quantification of the degree of synthesis between regional carbon sink capacity and high-quality development level, under the influence of both nature and society, is helpful in comprehensively understanding the pattern and mechanism of carbon sink potential in Yunnan Province. The trend surface fitting results show that (1) the spatial trend line of the comprehensive index of carbon sink capacity maintained the layout of "high in the west and low in the east, high in the north and south poles and low in the middle", showing obvious spatial directivity, with the west, north, and south being the main areas of carbon sinks (Figure 7a). (2) The high-quality development index showed a trend line difference of "low east-west, low north-south, high in the middle", which means that the high-quality development level is more prominent and stronger in central Yunnan, with agglomeration as the core area and radiating outward (Figure 7b).

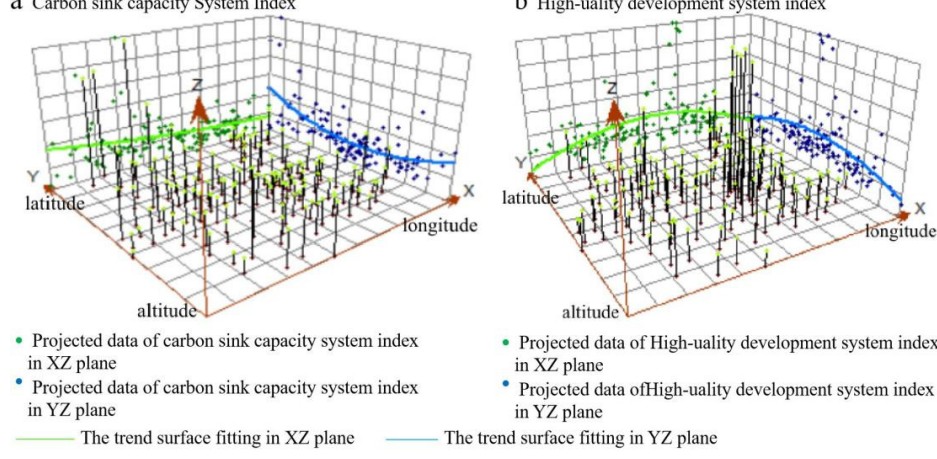

**Figure 7.** The trend surface fitting analysis: (**a**) carbon sink capacity system index; (**b**) high-quality development system index.

### 5.2.2. Analysis of Coupling Coordination between Carbon Sink Capacity and High-Quality Development

The coupling coordination level between carbon sink capacity and high-quality development range from 0.18~0.74. Based on the existing research results and calculation results, the coupling coordination types were divided into five categories, including unbalanced coordination, primary coordination, and intermediate coordination [35]. As shown in Figure 8a, the overall spatial structure of the provincial coordination level is discrete, with the following characteristics: (1) The high-quality and well-coordinated regions have a stable spatial structure with a coordinated mutual relationship, mainly located in the south of Kunming, the west of Yuxi, the east of Pu'er, the middle of Dali, and the northwest of Nujiang. The central counties are the single core of high value, and the surrounding counties are the secondary core. (2) The intermediate coordination and primary coordination areas are mainly distributed in most areas of Baoshan, Qujing, and Honghe. Driven by the high level of regional radiation, high-quality development is enhanced, and the integration degree with the carbon sink capacity is strengthened. Therefore, the intermediate coordination and primary coordination areas are open surrounding outside the high-value areas, and promote the convergence of spatial functional coordination levels among central counties. (3) The unbalanced coordination areas are mainly distributed in the west and east of Honghe, the south of Wenshan, the central and east of Chuxiong, the central and east of Zhaotong, and the west of Diqing. These regions present a low-value core due to the relatively sparse distribution and small volume of economic entity and living space entity distribution, and the influence of complex terrain, water, agriculture, forestry, and other natural ecological elements.

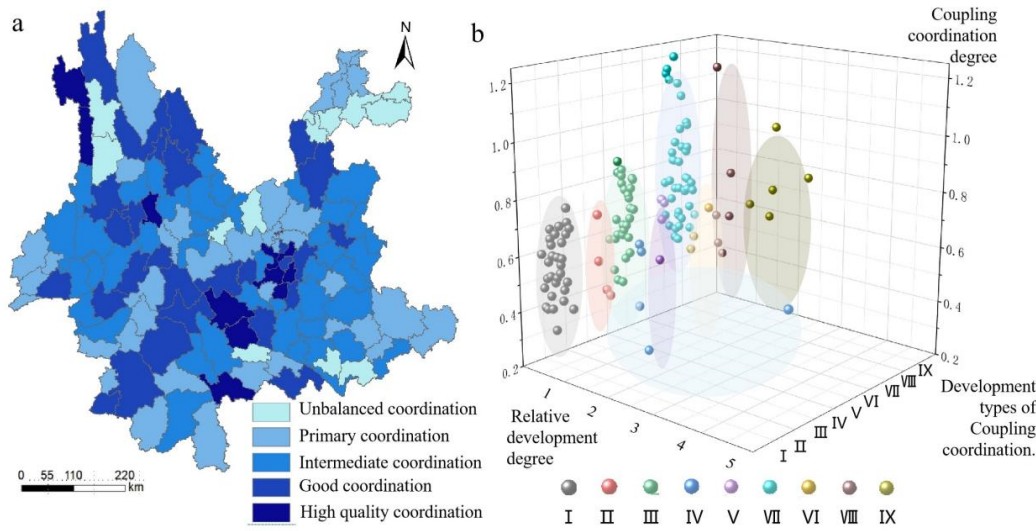

**Figure 8.** Coupling coordination degree between carbon sink capacity and quality development: (**a**) degree of coupling coordination between carbon sink capacity and quality development; (**b**) development types of coupling coordination.

According to the calculation results of the coupling coordination degree and the basis for judging the coupling coordination development stage in Table 4, the coupling coordination development stage between the spatial carbon sink capacity and high-quality development of counties in Yunnan Province can be evaluated (Figure 8). (1) The overall development stage of coupling coordination degree shows a trend of "wide at both ends and narrow in the middle". Specifically, in 128 counties, the coupling coordination degree is mostly in the antagonistic stage and coordination stage, accounting for 31% and 41% of the total number of counties, respectively, and 28% in the coordination stage. At present, the period of high-quality economic and social development is synchronized, and the promotion of carbon sink capacity inevitably leads to land-use contradictions in many aspects. Identifying these key factors is an important part of the process of promoting carbon seques-

tration capacity and high-quality development. (2) Generally speaking, the characteristic of coupling and coordinated development types shows that carbon sink in most counties lags behind and restricts high-quality development. In the areas that the system tends to decline, degradation, and optimization present an "inverted triangle" situation. That is, there are fewer regions in "synchronous state" and "advanced state", which shows that ecological carbon sink capacity is weak in most areas. Investigating its reasons, there are the following aspects: firstly, at the present stage, our country is still in the mode of economic development of sustained high-speed growth. The past 40 to 50 years of development have been driven by intensive resource exploitation and high-energy consumption. The capacity and potential of ecological carbon sinks need to be further enhanced. Secondly, most counties in Yunnan Province have extensive economic development modes. In the process of local governance, along with the rapid development of urbanization, the development concept of economic supremacy and the extensive economic growth model have been formed, as well as the blind pursuit of urbanization rate and land finance, the ecological carbon sink space within the region is crowded out. Coupled with the insufficient investment in protection and rehabilitation, inefficient use of energy and resources, and ineffective control of pollutant emissions, this ultimately restricts the capacity of carbon sinks and makes high-quality development difficult to reflect. Finally, the lack of natural resource management assessment mechanism and the lagging construction of ecological carbon sink protection system hinder the improvement of ecological carbon sink capacity and the development of a green economy. It should be noted that no matter if the region is "ahead" or "behind", it is not conducive to the coupling of carbon sink capacity to high-quality sustainable development. Regions that are ahead should devote themselves to improving the quality of their economic development. Regions that are lagging should improve the level of ecological carbon sink, and give full play to the efficiency of carbon sink in important ecological carbon sequestration land to promote the coupling and coordination of carbon sequestration capacity and high-quality development.

**Table 4.** Phases and types of coupling coordination development between ecological civilization and urbanization.

| Degree of Coupling Coordination Development | Relative Development Degree | Type | Coupling Coordinated Development Type Characteristics | Development Phase | Counties |
|---|---|---|---|---|---|
| $0 \leq D \leq 0.35$ | $0 < P \leq 0.8$ | I | The carbon sink function lags behind the high-quality development and restricts the high-quality development, the ecological carbon sink capacity is weak, and the system tends to decline. | Antagonistic stage | Wuhua, Shuifu, Hongta, Suijiang, Ruili City, Lianghe, Ximeng, Xichou, Weixin, Fumin, Midu, Luxi, Mouding, Ludian, Daguan, Yanchin, Zhaoyang, Pingbian, Yun County, Honghe, Cangyuan, Yiliang, Shuangjiang, Yuanyang, Maguan, Lufeng, Xundian, Yongshan, Lvchun, Linxiang, Wu Ding xian |
| | $0.8 < P < 1.2$ | II | The carbon sink function synchronizes high-quality development and promotes high-quality development, the ecological carbon sink capacity is enhanced, and the system tends to be optimized. | | Chuxiong, Funing, Lanping, Weixi |
| | $1.2 \leq P$ | III | The carbon sink function is ahead of the high-quality development, which affects the high-quality development, the high-quality development lags behind, and the system tends to be degraded. | | Yingjiang, Mengla, Guangnan, Zhenxiong, Shangri-La |

**Table 4.** *Cont.*

| Degree of Coupling Coordination Development | Relative Development Degree | Type | Coupling Coordinated Development Type Characteristics | Development Phase | Counties |
|---|---|---|---|---|---|
| 0.35 < D < 0.5 | 0 < P ≤ 0.8 | IV | The carbon sink function lags behind the high-quality development and restricts the high-quality development, the ecological carbon sink capacity is weak and the system tends to decline. | Breaking-in stage | Anning, Songming, Malone, Shidian, Longchuan, Shilin, Huaping, Yao 'an, Eryuan, Weishan, Luoping, Heqing, Mengzi, Jianchuan, Zhenkang, Eshan, Shizong, Nanhua, Wenshan, Malipo, Longling, Yun, Yanshan, Simao, Fuyuan, Meili, Gengma, Changning |
| | 0.8 < P < 1.2 | V | The carbon sink function synchronizes high-quality development and promotes high-quality development, the ecological carbon sink capacity is enhanced, and the system tends to be optimized. | | Menghai, Tengchong, Xuanwei, Qiubei, Mojiang |
| | 1.2 ≤ P | VI | The carbon sink function is ahead of the high-quality development, which affects the high-quality development, the high-quality development lags behind, and the system tends to be degraded. | | Lushui, Luquan, Longyang |
| 0.5 ≤ D ≤ 1 | 0 < P ≤ 0.8 | VII | The carbon sink function lags behind the high-quality development and restricts the high-quality development, the ecological carbon sink capacity is weak, and the system tends to decline. | Coordination stage | Panlong, Chenggong, Guandu, Chengjiang, Tonghai, Xishan, Jiangchuan, Qilin District, Hekou, Menglian, Yuanjiang, Huating, Dali, Luliang, Yimen |
| | 0.8 < P < 1.2 | VIII | The carbon sink function synchronizes high-quality development and promotes high-quality development, the ecological carbon sink capacity is enhanced, and the system tends to be optimized. | | Dayao, Fugong, Yongsheng, Jinghong, Ninglang, Huize |
| | 1.2 ≤ P | IX | The carbon sink function is ahead of the high-quality development, which affects the high-quality development, the high-quality development lags behind, and the system tends to be degraded. | | Jinggu, Lancang, Yulong, Gongshan, Deqin |

*5.3. Analysis of Influencing Factors*

5.3.1. Single-Factor Detection

To further explore the main factors influencing carbon sink capacity, high-quality development, coupling coordination degree, and the interaction of different factors, the influence factors were sorted according to differences and the *q*-values of factor detection results [38] (Table 5). The specific characteristics are as follows:

**Table 5.** Single-factor detection result.

| Comprehensive Index of Carbon Sink Capacity | | | Comprehensive Index of High-Quality Development | | | Coupling Coordination Degree | | |
|---|---|---|---|---|---|---|---|---|
| Factor Sorting | $q$ | $p$ | Factor Sorting | $q$ | $p$ | Factor Sorting | $q$ | $p$ |
| X1 | 0.013 | 0.554 * | X9 | 0.016 | 0.405 * | X9 | 0.054 | 0.315 |
| X6 | 0.001 | 0.503 * | X5 | 0.087 | 0.369 | X10 | 0.019 | 0.250 * |
| X2 | 0.028 | 0.410 * | X8 | 0.029 | 0.354 * | X4 | 0.041 | 0.247 * |
| X7 | 0.083 | 0.345 | X1 | 0.018 | 0.343 * | X12 | 0.059 | 0.186 |
| X8 | 0.042 | 0.223 * | X10 | 0.088 | 0.272 | X2 | 0.014 | 0.123 * |

Note: The $q$-values indicate that the influence factor explains $100 \times q\%$ of the dependent variable. The $q$-values marked with * indicate passing the significance test of $p \leq 0.05$.

(1) The core driving factors of the comprehensive index of carbon sink capacity are altitude (X1) > area of afforestation (X6) > slope (X2) > green space coverage of built-up area (X7) > land for water conservancy facilities (X8). This shows that terrain has a significant influence on carbon sink capacity, and vegetation restoration, and afforestation, and other measures promote the absorption of atmospheric carbon dioxide. Investing in infrastructure, such as through the construction of water conservancy facilities, improves the water conditions of farmland and regional water conservancy conditions, enhances the ability to resist natural disasters, and promotes a virtuous cycle of the ecological environment.

(2) The main influencing factors of the high-quality development index were river network density (X9) > regional population (X5) > land for water conservancy facilities (X8) > altitude (X1) > land for transportation (X10). It can be seen that the development and distribution of the river network plays an essential role in the local high-quality development of transportation, irrigation, power generation, aquatic products, ecology, and other areas. Secondly, population also has an impact on quality development in terms of labor supply and consumer demand. Thirdly, based on the topography and landform of the southwest plateau, slope and elevation affect the regional economic benefits and development potential by influencing land use. Finally, the exploitation of transportation land in Yunnan could form the system construction of economic and logistics nodes, boost the exploitation of resources, tourism, and industrial construction, and even drive the development of rural areas.

(3) The dominant driving factors of coupling coordination degree were river network density (X9) > road network density (X10) > soil type (X4) > night light index (X12) > slope (X2). This shows that the variation of carbon sink capacity and high-quality development coupling coordination degree in Yunnan Province mainly depends on regional natural background conditions, economic development, and urbanization level.

5.3.2. Interactive Factor Detection

The results of the interaction factor detection indicate that the carbon sink capacity, high-quality development, and their coupling degree in Yunnan Province are influenced by the combined action of multiple factors with double-factor enhancement and nonlinear enhancement relationships.

(1) Figure 9a shows that water facility land (X8) and elevation (X1) have the strongest interaction factors with the composite index of carbon sink capacity, followed by road network density (X10), soil type (X4), plantation area (X6), and green area coverage of built-up areas (X7), with $q$-values above 0.56.

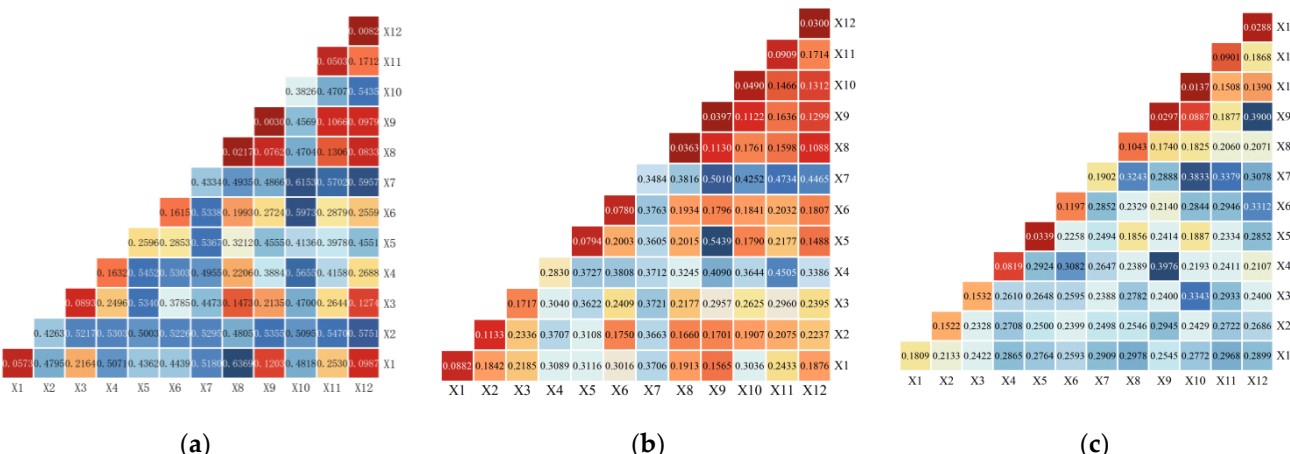

**Figure 9.** Two-factor interaction detection results of spatial-temporal variation of coupling coordination degree of carbon sink capacity and high-quality development in Yunnan Province: (**a**) the composite index of carbon sink capacity; (**b**) the composite index of high-quality development; (**c**) coupling coordination degree.

(2) The main interaction factors of the composite index of high-quality development are presented in Figure 9b, which indicates a strong interaction between the density of the river network (X9) and the regional population (X5), and the green area coverage of the built-up area (X7), with *q*-values above 0.5.

(3) Figure 9c illustrates that the main interaction factors of coupling coordination are the river network density (X9) and soil quality (X4), which show a strong interaction with the nighttime light index (X12), with *q*-values above 0.39. However, the overall interaction showed a weakening trend. The interaction results further verified that the variation of coupling coordination degree of carbon sink capacity and high-quality development in Yunnan Province mainly depend on regional natural background conditions, economic development, and urbanization level.

### 5.3.3. Influence Mechanism Analysis

The natural environment provides the foundation for the utilization and protection of national space, and is a prerequisite for achieving high-quality development. The carbon sink capacity is primarily affected by regional background conditions, and is supported by conservation and restoration initiatives. High-quality development is achieved through the construction of an ecological security pattern with rivers and lakes as important ecological corridors, under appropriate slope, altitude, and exploitation conditions. This includes improving the transportation development system, realizing the pattern of economic development, and supporting the highly unified and sustainable development of ecology and economy. The coupling and coordination relationship is the representation of ecological protection and the stage of the urbanization process [39]. In the process of urbanization, regional economic development is guaranteed and labor force is effectively attracted based on development needs. In the process of land and space development, the efficiency of natural resources and function of social security will be brought into full play. The improvement of carbon sink capacity corresponds to the ecological service capacity provided by water conservation, biodiversity maintenance and carbon sequestration and oxygen release, and it also plays a positive feedback role for economic development and social security function.

## 6. Recommendations

Based on our research, we suggest that Yunnan Province focuses on the following aspects to improve its carbon sink capacity and high-quality development, based on the spatial exploitation and protection pattern.

A new pattern of national spatial exploitation and conservation should be constructed, emphasizing the coordination of carbon sink capacity and high-quality development. As Yunnan Province is situated on the border of southwest China, it has a complex landscape, fragile ecology, and a large poverty surface. The problems of unbalanced development between urban and rural areas, mountainous and dam areas, the border and the interior, and different ethnic groups are increasingly prominent. As an important carrier of urbanization construction, counties should first change their development mode in the development process. They should consider the corresponding stage of coupled and coordinated level of carbon sink capacity and high-quality development in each county, as well as the characteristics of coupled and coordinated development types (Table 4). Additionally, it is crucial to rationalize the size, population density, and spatial structure of counties and regions, build an industrial system that meets the region's background conditions, and promote efficient and concentrated economic development in a more balanced, healthy, and sustainable direction.

The ecological carbon sink capacity based on space and important ecological CSL should be enhanced [40]. Counties and districts must ensure that the amount of natural ecological space is not reduced. This can be achieved by increasing the proportion of important ecological CSL to improve the quality and stability of the ecological carbon sink system. Effective ecological restoration and protection require adherence to the mountain, water, forest, field, lake, and grass management system. Furthermore, strengthening the implementation of the river and lake system and forestry system, scientifically promoting the comprehensive management of desertification, stone desertification, soil erosion, and construction of national parks as the main nature reserve system are essential initiatives. The continuous adjustment and optimization of the spatial structure of ecological CSL will enable the optimization of the ecological carbon sink system, which serves as the basis for protection effectiveness, monitoring, and evaluation [41].

Establishing a mechanism for realizing the value of ecological carbon sinks is crucial, as their value is increasingly evident over time [42]. Yunnan Province has natural advantages for carbon sequestration and is well positioned to promote the transformation of ecological carbon sink value into economic value. This approach can facilitate high-quality economic development and high-level ecological carbon sink protection, in line with the principle of "green water and green mountain is the silver mountain." By integrating temporal and spatial dimensions, it is possible to achieve unified and coordinated development that balances economic, social, and ecological benefits.

**Author Contributions:** Conceptualization, J.Z.; Methodology, L.W.; Formal analysis, G.C.; Data curation, F.L.; Writing—original draft, L.W. All authors have read and agreed to the published version of the manuscript.

**Funding:** This research was funded by the Foundation of Yunnan Office of Philosophy and Social Science (YB2021015); National Science Foundation of China (417610815); curriculum ideological and political construction key subject in Kunming University of science and technology (2021KS004).

**Institutional Review Board Statement:** Not applicable.

**Informed Consent Statement:** Not applicable.

**Data Availability Statement:** Not applicable.

**Conflicts of Interest:** The authors declare no conflict of interest.

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
