# Peer review of "Spatial Coupling of Carbon Sink Capacity with High-Quality Development Based on Exploitation and Protection Pattern"

_sustainability, doi:10.3390/su15108108_

Round 1

Reviewer 1 Report (Previous Reviewer 1)

I'm satisfied with the revised and resubmitted version and endorsed the publication of this manuscript.

Minor spell check required.

Author Response

Response to the Academic Editor and Reviewers

Dear Editor(s),

Thank you for the valuable feedback on our manuscript titled "Spatial coupling of carbon sink capacity with high quality development based on exploitation and protection pattern"(Manuscript ID: Sustainability-2378405). We appreciate the time and effort invested in evaluating our work and we are grateful for the insightful comments.

We have carefully revised our manuscript following the comments and suggestions. We have used the track-change mode in the revised manuscript.

We enlisted a colleague who is good at English to review our writing, and revisions are made carefully. We hope that the revisions have met the requirements of Sustainability.

Very sincerely yours,

Lin Wang

The following is the changes made in response to the Academic Editor and reviewers' comments.

Academic Editor Comments

Point 1. Check the language carefully. There was an extra word "Global" in the first paragraph of the introduction.

Response 1:

We have conducted a comprehensive r proofreading of the manuscript, removing unnecessary words (including the extra word "Global”), correcting grammatical and typo errors.

Point 2. Figure 6 is missing the pointer and scale elements.

Response 2:

We have added the pointer and scale elements to Figure 6 and made sure that they are clearly displayed. (Line 85)

Reviewer 1

Minor editing of English language required

Minor spell check required.

Response 3:

We have carefully reviewed and revised the manuscript to improve the English writing. We also enhanced and optimized the clarity of the academic writing in particular. The track-change mode in Word has been used to mark the changes in the manuscript.

Reviewer 2

Comments and Suggestions for Authors:

The article, in my opinion is publishable in this form.

Response 4:

Many thanks for the constructive comments and valuable suggestions of the reviewer. We will continue to conduct in-depth research in related fields and strive for continuous improvement in our academic writing.

Sustainability Editorial Office

Please check that all references are relevant to the contents of the manuscript.

Response 5:

We have carefully reviewed our manuscript to ensure that all references are relevant to the contents of the manuscript, and we have revised the reference formatting to conform to Sustainability.

Reviewer 2 Report (Previous Reviewer 2)

The article, in my opinion is publishable in this form.

Author Response

Response to the Academic Editor and Reviewers

Dear Editor(s),

Thank you for the valuable feedback on our manuscript titled "Spatial coupling of carbon sink capacity with high quality development based on exploitation and protection pattern"(Manuscript ID: Sustainability-2378405). We appreciate the time and effort invested in evaluating our work and we are grateful for the insightful comments.

We have carefully revised our manuscript following the comments and suggestions. We have used the track-change mode in the revised manuscript.

We enlisted a colleague who is good at English to review our writing, and revisions are made carefully. We hope that the revisions have met the requirements of Sustainability.

Very sincerely yours,

Lin Wang

The following is the changes made in response to the Academic Editor and reviewers' comments.

Academic Editor Comments

Point 1. Check the language carefully. There was an extra word "Global" in the first paragraph of the introduction.

Response 1:

We have conducted a comprehensive r proofreading of the manuscript, removing unnecessary words (including the extra word "Global”), correcting grammatical and typo errors.

Point 2. Figure 6 is missing the pointer and scale elements.

Response 2:

We have added the pointer and scale elements to Figure 6 and made sure that they are clearly displayed. (Line 85)

Reviewer 1

Minor editing of English language required

Minor spell check required.

Response 3:

We have carefully reviewed and revised the manuscript to improve the English writing. We also enhanced and optimized the clarity of the academic writing in particular. The track-change mode in Word has been used to mark the changes in the manuscript.

Reviewer 2

Comments and Suggestions for Authors:

The article, in my opinion is publishable in this form.

Response 4:

Many thanks for the constructive comments and valuable suggestions of the reviewer. We will continue to conduct in-depth research in related fields and strive for continuous improvement in our academic writing.

Sustainability Editorial Office

Please check that all references are relevant to the contents of the manuscript.

Response 5:

We have carefully reviewed our manuscript to ensure that all references are relevant to the contents of the manuscript, and we have revised the reference formatting to conform to Sustainability.

This manuscript is a resubmission of an earlier submission. The following is a list of the peer review reports and author responses from that submission.

Round 1

Reviewer 1 Report

The manuscript “Spatial coupling of carbon sink capacity with high quality development based on exploitation and protection pattern” is interesting and under the scope of the journal ‘FORESTS.’ However, the Abstract and Introduction need improvement. Briefly, the MS needs minor revisions before possible acceptance in the ‘FORESTS’ journal.

Title

1-      Line 2-3: The title is good.

Abstract

1-      Line 12-32: According to journal (Forests) policy, the abstract should not be more than 200 words, and your abstract is more than 250. So please rewrite the precise and short Abstract to meet the journal’s requirements and for easy understanding of readers.

2-      Line 13: Please add space after ‘land-use

3-      Line 18: Please add space before ‘The spatial pattern

4-      Line 21: Please add space before ‘The development stage

5-      Line 23: Please add space before ‘Carbon sink capacity

6-      The space issue in the whole manuscript, please carefully check and address this issue.

1-      Line 31-32: Please remove the words (exploitation and protection; carbon sink capacity; high-quality development;) from the keywords because you have already used this word in the title, and add a more suitable word to enhance the visibility of your Article.

Introduction

Please incorporate this information into the Introduction:

1-      What scientific questions have you addressed?

2-      What will benefit the scientific community/society of your study?

3-      What is the significance of your research?

4-      A relevant hypothesis for the study is missing from the introduction. Please clarify your hypothesis.

5-      The authors are encouraged to consult and cite recent literature in the introduction.

https://doi.org/10.1016/j.catena.2022.106280, Catena

http://dx.doi.org/10.1016/j.scitotenv.2022.158610, Science of The Total Environment.

Materials and Methods

1-      The theoretical framework is excellent, and I appreciated the author’s efforts.

2-      The legend of Figure 2 should be comprehensive.

Hopefully, these suggestions will help you to improve your Article.

Good luck!

Reviewer 2 Report

Title:

Spatial coupling of carbon sink capacity with high quality development based on exploitation and protection pattern

Lin Wang, Junsan Zhao, Fengxia, Guoping Chen

The paper deal with a very actual subject of nowadays in the world: the authors  established a new the  spatial pattern and supporting system framework of land-use of carbon sink efficiency, measured  the coordination level between carbon sink capacity and high-quality development, and explored  the driving factors of carbon sink capacity coupling high-quality development level in Yunnan Province of southwestern China.

Abstract

Please structure the abstract as:

Introduction-Aims

Method

Results and interpretation

Introduction:

Please develop the literature review and update to 2023

Please analyses critically the findings of the articles and the limitations.

Please indicate also at least tree similar article to your research published recently (last 5 years).

Aim of the study is not clear indicated

Fig 2 line 120. Please numbering a,b,c and insert explanations to each one.

Legend to be revised to the left figure

One of its are not lisible

Methodology

Flowchart of the methodology steps to be inserted

Eq 1,2,3 pages 4 and 5 : please insert the authors

Eq 4 idem as above. Indicate also the page

Idem eq 6,7,8,9

To eq 8 please indicate each component what represents

Fig.3 a, b line 253: please indicate the unit for left figure for vertical axis. Indicate the datasource

Fig. 4 line 275: please indicate the units to the legends and datasource

Fig.5 line 304: please indicate the units to the legends and datasource

Fig 6. please indicate the units to the legends and datasource

Fig.7 line 345: it is missing the legend. The figs are ilisible.

Fig.8. insert also the datasource

References : must be developed and updated to 2023

We consider useful also https://doi.org/10.2478/jaes-2020-0005

What is the main question addressed by the research?

The main question addressed. The subject to which the paper address is very actual one, regarding spatial coupling of carbon sink capacity; the references are not enough exploited in a special subchapter of literature review. Please revise

Is it relevant and
interesting?

The paper is relevant especially nowadays in the complex environment of Yunnan  Province China, concerning the raise to land use carbon sink efficiency, facilitate coordination of carbon sink capacity and provides a new perspective for authorities to  ecological  exploitation and protection

How original is the topic?

Is an actual topic with medium degree of originality; but it is important subject especially in nowadays period, authors findings can contribute successfully to investigate the combined effects of spatial coupling of carbon sink capacity.

What does it add to the subject area compared with other published material?

The paper is not very well documented because the authors cited over 30 scientific published articles; we suggested to develop and to be updated to 2023.

We consider useful for the paper also the following published article. Please see and cite it:

We consider useful also https://doi.org/10.2478/jaes-2020-0005

Is the paper well written?

The paper is well written. The quality of English translation is good.
Is the text clear and easy to read?

The text is well structured. Methodology and discussions are satisfactory.

Are the conclusions consistent with the evidence and arguments presented?

Yes, The results and discussions subchapters are concluding and consistent

Best regards,

March  2023
